# PyHIST: A Histological Image Segmentation Tool

**Manuel Muñoz-Aguirre**[1,2⚬]*, **Vasilis F. Ntasis**[1⚬], **Santiago Rojas**[3], **Roderic Guigó**[1,4]

**1** Centre for Genomic Regulation (CRG), The Barcelona Institute of Science and Technology, Barcelona, Catalonia, Spain, **2** Department of Statistics and Operations Research, Universitat Politècnica de Catalunya (UPC), Barcelona, Catalonia, Spain, **3** Unit of Human Anatomy and Embryology. Department of Morphological Sciences. Faculty of Medicine. Universitat Autònoma de Barcelona, Cerdanyola del Vallès, Catalonia, Spain, **4** Department of Experimental and Health Sciences (DCEXS), Universitat Pompeu Fabra (UPF), Barcelona, Catalonia, Spain

⚬ These authors contributed equally to this work.
* manuel.munoz@crg.eu

**Data Availability Statement:** All relevant data are within the manuscript and its Supporting Information files.

**Funding:** The authors received no specific funding for this work. M.M.-A. performs his research with

## Abstract

The development of increasingly sophisticated methods to acquire high-resolution images has led to the generation of large collections of biomedical imaging data, including images of tissues and organs. Many of the current machine learning methods that aim to extract biological knowledge from histopathological images require several data preprocessing stages, creating an overhead before the proper analysis. Here we present PyHIST (https://github.com/manuel-munoz-aguirre/PyHIST), an easy-to-use, open source whole slide histological image tissue segmentation and preprocessing command-line tool aimed at tile generation for machine learning applications. From a given input image, the PyHIST pipeline i) optionally rescales the image to a different resolution, ii) produces a mask for the input image which separates the background from the tissue, and iii) generates individual image tiles with tissue content.

## Author summary

Histopathology images are an essential tool to assess and quantify tissue composition and its relationship to disease. The digitization of slides and the decreasing costs of computation and data storage have fueled the development of new computational methods, especially in the field of machine learning. These methods seek to make use of the histopathological patterns encoded in these slides with the aim of aiding clinicians in healthcare decision-making, as well as researchers in tissue biology. However, in order to prepare digital slides for usage in machine learning applications, researchers usually need to develop custom scripts from scratch in order to reshape the image data in a way that is suitable to train a model, slowing down the development process. With PyHIST, we provide a toolbox for researchers that work in the intersection of machine learning, biology and histology to effortlessly preprocess whole slide images into image tiles in a standardized manner for usage in external applications.

support of pre-doctoral fellowship FPU15/03635 from Ministerio de Educación, Cultura y Deporte. (URL: http://www.mecd.gob.es/) Agencia Estatal de Investigación (AEI) and FEDER under project PGC2018-094017-B-I00 is also acknowledged. The funders had no role in study design, data collection and analysis, decision to publish, or preparation of the manuscript.

**Competing interests:** The authors have declared that no competing interests exist.

This is a *PLOS Computational Biology* Software paper.

## Introduction

In histopathology, Whole Slide Images (WSI) are high-resolution images of tissue sections obtained by scanning conventional glass slides [1]. Currently, these glass slides of fixed tissue samples are the preferred method in pathology laboratories around the world to make clinical diagnoses [2], notably in cancer [3]. However, the increasing automation of WSI acquisition has led to the development of computational methods to process the images with the goal of helping clinicians and pathologists in diagnosis and disease classification [4]. As an increasing number of larger WSI datasets became available, methods have been developed for a wide array of tasks, such as the classification of breast cancer metastases, Gleason scoring for prostate cancer, tumor segmentation, nuclei detection and segmentation, bladder cancer diagnosis, mutated gene prediction, among others [5–10]. Besides of being important diagnostic tools, histopathological images capture endophenotypes (of organs and tissues) that, when correlated with molecular and cellular data on the one hand, and higher-order phenotypic traits on the other, can provide crucial information on the biological pathways that mediate between the sequence of the genome and the biological traits of the organisms (including diseases) [11].

Because of the complexity of the information typically contained in WSIs, Machine Learning (ML) methods that can infer, without prior assumptions, the relevant features that they encode are becoming the preferred analytical tools [12]. These features may be clinically relevant but challenging to spot even for expert pathologists, and thus, ML methods can prove valuable in healthcare decision-making [13].

In most ML tasks, data preprocessing remains a fundamental step. Indeed, in the domain of histological images, there are several issues when preprocessing the data before an analysis: due to the large dimensions of WSIs, many deep learning applications have to break them down into smaller-sized square pieces called tiles [14]. Furthermore, a significant fraction of the area in a WSI is often uninformative background that is not meaningful for the majority of downstream analyses. To circumvent this, some applications apply a series of image transformations to identify the foreground from the background (see, for example, [15]), and perform relevant operations only over regions with tissue content. However, this process is not standardized, and customized scripts have to be frequently developed to deal with data preparation stages (see, for example [10,15]). This is cumbersome and may introduce dataset specific-biases, which can prevent integration across multiple datasets.

Currently available tools for WSI processing focus mostly on the analysis of human-interpretable features by means of nuclei segmentation, object quantification and region-of-interest annotation [16–18]; but WSI preparation into tiles for external ML applications has not yet been directly addressed. To systematize the WSI preprocessing procedure for these applications, and in order to streamline the data preparation stage at the initial phase of a ML project by avoiding the need of creating custom image preprocessing scripts, we developed PyHIST, a command-line based pipeline to segment the regions of a histological image into tiles with relevant tissue content (foreground) with little user intervention. PyHIST was developed to process Aperio SVS/TIFF WSIs due to this format being supported by large slide databases such as The Cancer Genome Atlas (TCGA) which has approximately 31,000 WSIs [19] and The Genotype-Tissue Expression Project (GTEx) with approximately 25,000 WSIs [20]. PyHIST currently has experimental support for other image formats (see S1 Text).

## Design and implementation

PyHIST is a command-line Python tool based on OpenSlide [21], a library to read high-resolution histological images in a memory-efficient way. PyHIST's input is a WSI encoded in SVS format (Fig 1A), and the main output is a series of image tiles retrieved from regions with tissue content (Fig 1E).

The PyHIST pipeline involves three main steps: 1) produce a mask for the input WSI that differentiates the tissue from the background, 2) create a grid of tiles on top of the mask, evaluate each tile to see if it meets the minimum content threshold to be considered as foreground and 3) extract the selected tiles from the input WSI at the requested resolution. By default, PyHIST uses a graph-based segmentation method to produce the mask. In this method, first, tissue edges inside the WSI are identified using a Canny edge detector (Fig 1B), generating an alternative version of the image with diminished noise and an enhanced distinction between the background and the tissue foreground. Second, these edges are processed by a graph-based segmentation algorithm [22], which is used here to identify tissue content. In short, this step evaluates the boundaries between different regions of an image as defined by the edges;

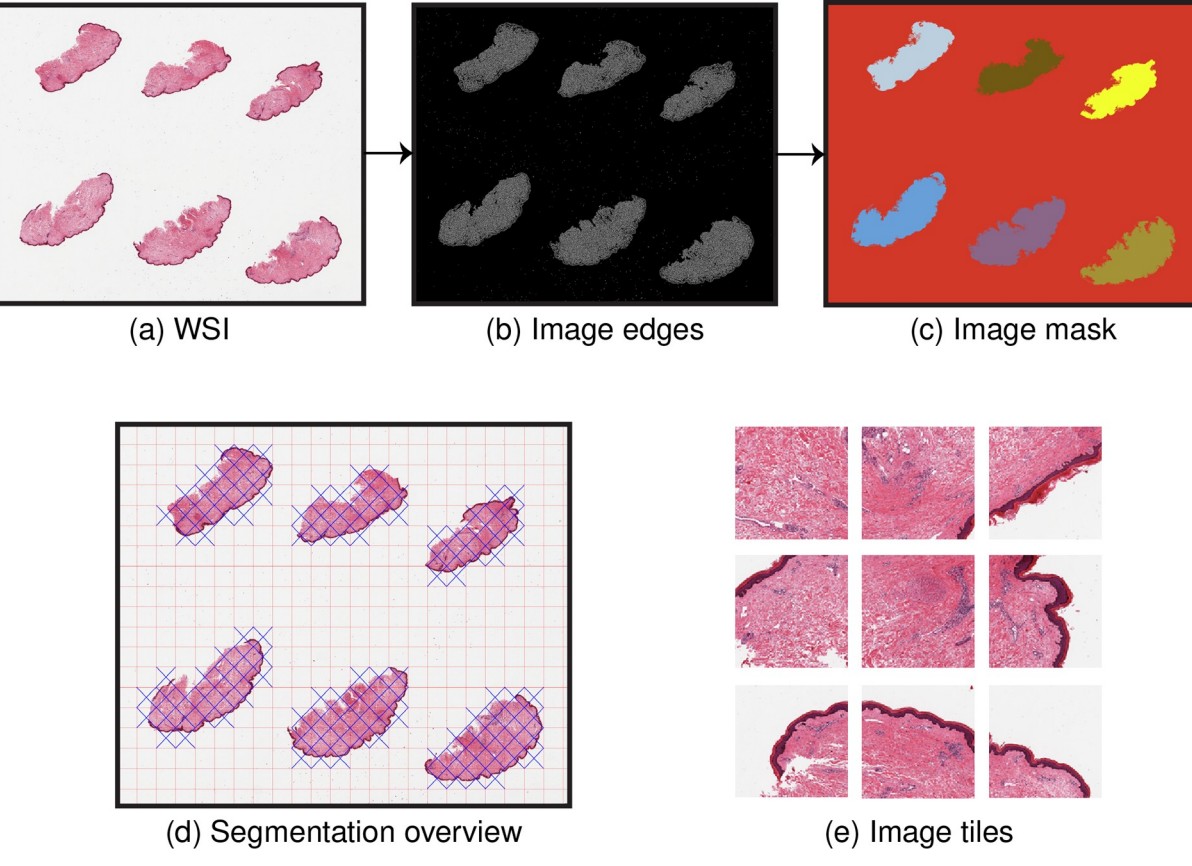

(a) WSI                (b) Image edges               (c) Image mask

(d) Segmentation overview              (e) Image tiles

**Fig 1. PyHIST pipeline.** (a) The input to the pipeline is a Whole Slide Image (WSI). Within PyHIST, the user can decide to scale down the image to perform the segmentation and tile extraction at lower resolutions. The WSI shown is of a skin tissue sample (GTEX-1117F-0126) from the Genotype-Tissue Expression (GTEx) project [20]. (b) An alternative version of the input image is generated, where the tissue edges are highlighted using a Canny edge detector. A graph segmentation algorithm is employed over this image in order to generate the mask shown in (c). PyHIST extracts tiles of specific dimensions from the masked regions, and provides an overview image to inspect the output of the segmentation and masking procedure, as shown in (d), where the red lines indicate the grid generated by tiling the image at user-specified tile dimensions, while the blue crosses indicate the selected tiles meeting a certain user-specified threshold of tissue content with respect to the total area of the tile. In (e), examples of selected tiles are shown.

different parts of the image are represented as connected components of a graph, and the "within" and "in-between" variations of neighboring components are assessed in order to decide if the examined image regions should be merged or not into a single component. From this, a mask is obtained in which the background and the different tissue slices are separated and marked as distinct objects using different colors (Fig 1C). Finally, the mask is divided into a tile grid with a user-specified tile size. These tiles are then assessed to see if they meet a minimum foreground (tissue) threshold with respect to the total area of the tile, in which case they are kept, and otherwise are discarded. Optionally, the user can also decide to save all the tiles in the image.

Of note, tile generation can be performed at the native resolution of the WSI, but downsampling factors can also be specified to generate tiles at lower resolutions. Additionally, edge detection and mask generation can also be performed on downsampled versions of WSIs—reducing segmentation runtimes (S1 Fig, S1 Text). A segmentation overview image is generated at the end of the segmentation procedure for the user to visually inspect the selected tiles (Fig 1D). With the set of parameters available in PyHIST (S2 Text), the user can specify regions to ignore when performing the masking and segmentation (S2 Fig), and have a fine-grained control over specific use-cases.

By default, PyHIST uses the graph-based segmentation method described previously due to its robustness in detecting tissue foreground in WSIs that do not have a homogeneous composition. However, alternative tile-generation methods based on thresholding that tend to work well on heterogeneous WSIs are also implemented (S3–S5 Figs, see S1 Text for details and benchmarking information). PyHIST also has a random tile sampling mode for those applications that do not necessarily need to distinguish the background from the foreground. In this mode, tiles at a user-specified size and resolution will be extracted from random starting positions in the WSI.

## Results

To demonstrate how PyHIST can be used to preprocess WSIs for usage in a ML application, we generated a use case example with the goal of building a classifier at the tile-level that allows us to determine the cancer-affected tissue of origin based on the histological patterns encoded in these tiles. To this end, we first retrieved a total of 36 publicly available WSIs, six from each of the following human tissues hosted in The Cancer Genome Atlas (TCGA) [23]: Brain (glioblastoma), Breast (infiltrating ductal carcinoma), Colon (adenocarcinoma), Kidney (clear cell carcinoma), Liver (hepatocellular carcinoma), and Skin (malignant melanoma). Slides within each tissue have the same cancer primary diagnosis as established by TCGA. Second, these WSIs were preprocessed with PyHIST, generating a total of 7163 tiles with dimensions 512x512. These tiles were then partitioned into training and test sets (constraining all the tiles of a given WSI to be in only one of the two sets), and we then fit a deep learning convolutional neural network model over these tiles with weighted sampling at training time (S6 Fig), achieving a classification accuracy of 95% (Fig 2A, S1 Table, S2 Table, see S3 Text for data preparation and model details, and a detailed assessment of Fig 2A).

We also inspected the feature vectors generated by the deep learning model: for each tile, we retrieved the features corresponding to the linear layer of the last (fully connected) sequential container of the model, and performed dimensionality reduction (t-SNE) over the stacked matrix of these vectors. From here, we infer that the learned features recapitulate tissue morphology since tile clusters corresponding to each tissue are formed (Fig 2B, S7 Fig). We note that this classifier is only an exercise to show end-users how to quickly prepare WSI data using PyHIST to generate tiles, reducing the overhead to start performing downstream analyses:

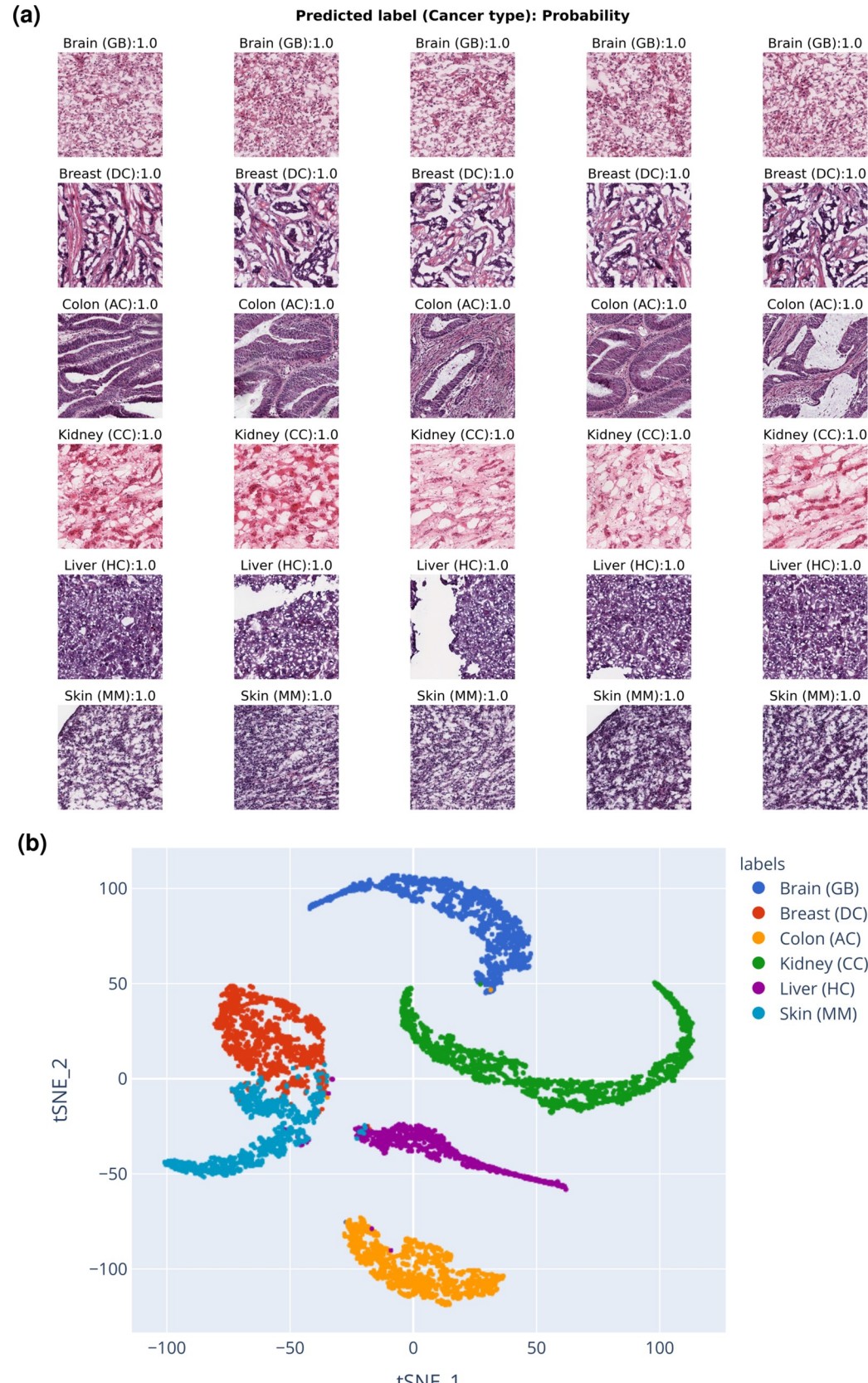

**Fig 2. TCGA use case.** (a) Examples of the top 5 most accurately predicted tiles per cancer-affected tissue (rows) from the TCGA use case test set. The label above each tile shows the predicted cancer-affected tissue type (GB: glioblastoma, DC: infiltrating ductal carcinoma, AC: adenocarcinoma, CC: clear cell carcinoma, HC: hepatocellular carcinoma, MM: malignant melanoma), followed by the probability of the ground truth label. All of these tiles were correctly classified. (b) Dimensionality reduction of TCGA tiles. t-SNE performed with the feature vectors of each tile that were derived from the deep learning classifier model. Each dot corresponds to an image tile.

further tuning of the model with more data is desirable to ensure that the classifier is robust enough to generalize to different types of unseen WSIs for a real application.

## Availability and future directions

The example use case described above is documented and fully available at https://pyhist. readthedocs.io/en/latest/testcase/, and divided into three Jupyter notebooks: 1) Data preprocessing with PyHIST, 2) Constructing a deep learning tissue classifier, and 3) Dimensionality reduction. The TCGA WSIs in the use case were downloaded from the Genomic Data Commons (GDC) repository (https://gdc.cancer.gov/) using the GDC Data-transfer tool (https:// gdc.cancer.gov/access-data/gdc-data-transfer-tool).

PyHIST is a generic tool to segment histological images automatically: it allows for easy and rapid WSI cleaning and preprocessing with minimal effort to generate image tiles geared towards usage in ML analyses. The tool is available at https://github.com/manuel-munoz-aguirre/PyHIST and released under a GPL license. Updated documentation and a tutorial can be found at https://pyhist.readthedocs.io/. PyHIST is highly customizable, enabling the user to tune the segmentation process in order to suit the needs of any particular application that relies on histological image tiles. The software and all of its dependencies have been packaged in a Docker image, ensuring portability across different systems. PyHIST can also be used locally within a regular computing environment with minimal requirements. Future directions and improvements include adding support for more histological image formats and features to save tiles into specialized data structures, as well as the inclusion of a graphical user interface to ease the learning curve for users who are new to the field of image processing for ML analyses. Finally, PyHIST is open source software: all the code and reproducible notebooks for the example use case are available in GitHub and will continue to be improved based on user feedback.

## Supporting information

**S1 Text. PyHIST overview.** General description of the pipeline: supported file formats, tile generation methods, and execution times.
(PDF)

**S2 Text. Parameter description.** Description of supported arguments in PyHIST.
(PDF)

**S3 Text. TCGA tissue classification use case.** Description of data preprocessing, model training and analysis for the TCGA tissue classification use case.
(PDF)

**S1 Fig. WSI scaling steps in PyHIST.** (a) WSI at its original resolution (1x). (b) The mask can be generated and processed at a given downsampling factor. A smaller resolution will lead to a faster segmentation. (c) The output can be requested at a given downsampling factor. (d) The segmentation overview image can also be generated at a given downsampling factor. The dimensions in all steps are matched to ensure that the tile sizes and grid are consistent. The

downsampling choices for all the steps are independent of each other.
(PNG)

**S2 Fig. Image in graph-based segmentation test mode.** Test mode allows the user to see how the image mask will be with the chosen segmentation parameters and tile dimension configuration, before proceeding to generate the individual tile files. The black border defines the region of exclusion for tissue content placed within the edges of the slide (see—*borders* and—*corners* arguments, and section 2.2 in S2 Text).
(PNG)

**S3 Fig. Comparison of mask generation methods.** (a) Adipose tissue WSI from the GTEx project, from sample GTEX-111CU-1826. Thresholding-based masks (b-d) are generated by first converting (a) into grayscale and then applying the corresponding thresholding method. Note that simple thresholding is shown here for completeness but only Otsu and adaptive are implemented in PyHIST due to their overall better performance when compared to simple thresholding. In the graph-based method, an image with highlighted edges is first generated through a Canny edge detector (e, left) and then the connected components are labeled through graph-based segmentation (e, right).
(PNG)

**S4 Fig. Runtime benchmarks for random sampling and graph-based segmentation.** (a) Execution time to perform random sampling (y-axis) of a varying number of tiles (x-axis) at different downsampling factors for the WSI shown in S1 Fig. For each combination of number of tiles and downsampling factor, the sampling was repeated 30 times. Each dot represents the average running time across the 30 runs, while the interval shows the range between the maximal and minimal running time. (b) Execution time to perform random sampling of 1000 tiles (y-axis) at different tile dimensions (x-axis) at different downsampling factors for the same WSI in (a). Each combination was repeated 50 times, with each dot showing the average runtime. (c) Segmentation runtime of 50 Stomach WSIs from the GTEx project, at different downsampling factors, at a tile size of 256x256. Each dot represents the average execution time. Each interval shows the range between the fastest and slowest segmentations, while the labels show the dimensions of the corresponding WSIs. (d) Segmentation runtime (y-axis) at 1x resolution for the 50 Stomach WSIs, with respect to the number of pixels in the WSI (x-axis).
(PNG)

**S5 Fig. Runtime comparison of mask-generating methods.** Tile extraction was evaluated for the three different methods at four different settings of tile size. Each method + tile size combination was repeated ten times to show runtime variability.
(PNG)

**S6 Fig. Tile distribution per class in a training epoch in the TCGA example use case.** Within each training epoch, weighted random sampling is performed to create batches with a fair distribution of tiles among the classes. Even if the sample sizes in the training dataset are different among the classes, the balance in the number of tiles per epoch is obtained through data augmentation.
(PNG)

**S7 Fig. Correlation matrix of TCGA tiles based on their feature vectors.** Heatmap of Pearson's correlation matrix between the feature vectors obtained for each TCGA tile. Rows and columns are reordered with hierarchical agglomerative clustering.
(PNG)

**S1 Table. Tile distribution across classes in the TCGA use case training and test sets.**
(PNG)

**S2 Table. Confusion matrix for the tiles in the test set of the TCGA use case.**
(PNG)

## Acknowledgments

We acknowledge Kaiser Co and Valentin Wucher for testing PyHIST, and the colleagues at the lab for useful feedback; Ferran Marqués, Verónica Vilaplana and Marc Combalia for useful discussions about image processing. All authors acknowledge the support of the Spanish Ministry of Science, Innovation and Universities to the EMBL partnership, the Centro de Excelencia Severo Ochoa, and the CERCA Programme / Generalitat de Catalunya.

## Author Contributions

**Conceptualization:** Manuel Muñoz-Aguirre, Vasilis F. Ntasis, Roderic Guigó.

**Formal analysis:** Manuel Muñoz-Aguirre, Vasilis F. Ntasis.

**Methodology:** Manuel Muñoz-Aguirre, Vasilis F. Ntasis.

**Software:** Manuel Muñoz-Aguirre, Vasilis F. Ntasis.

**Supervision:** Manuel Muñoz-Aguirre, Roderic Guigó.

**Validation:** Manuel Muñoz-Aguirre, Vasilis F. Ntasis, Santiago Rojas.

**Writing – original draft:** Manuel Muñoz-Aguirre, Vasilis F. Ntasis.

**Writing – review & editing:** Manuel Muñoz-Aguirre, Vasilis F. Ntasis, Santiago Rojas, Roderic Guigó.

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
