## [Decision Letter · Decision Letter 0]

17 Jul 2020

Dear Muñoz-Aguirre,

Thank you very much for submitting your manuscript "PyHIST: A Histological Image Segmentation Tool" for consideration at PLOS Computational Biology.

As with all papers reviewed by the journal, your manuscript was reviewed by members of the editorial board and by several independent reviewers. In light of the reviews (below this email), we would like to invite the resubmission of a significantly-revised version that takes into account the reviewers' comments.

We cannot make any decision about publication until we have seen the revised manuscript and your response to the reviewers' comments. Your revised manuscript is also likely to be sent to reviewers for further evaluation.

Sincerely,

Dina Schneidman-Duhovny

Software Editor

PLOS Computational Biology

Dina Schneidman-Duhovny

Software Editor

PLOS Computational Biology

Reviewer's Responses to Questions

**Comments to the Authors:**

Reviewer #1: The paper introduces a simple and easy to use open-source tool for slide tiling, which is useful for histopathology image analysis. I have to acknowledge there are currently inadequate tools available that make tiling of digital slides simple, and I had to write my own scripts to tile my slides when analyzing whole slide images. Overall, my experience with the application was positive; I was able to create tiles from svs and TIFF files in a short amount of time, with a short learning curve.

The setup and installation was relatively easy; the Docker version of the software worked without problems on Ubuntu and Windows 10 based machines. A few problems were encountered when the program was installed through Anaconda using the program’s accompanying installation instructions: in the Ubuntu machine, ‘cv2’ was reported to be missing; in Windows 10, several libraries were missing. I was able to get the program running in Ubuntu and Anaconda after manually installing OpenCV. I did not attempt to manually fix the Anaconda installation in Windows due to the large number of missing libraries. I am not sure if this is due to the unique setup of my computer, or the program was not tested with Windows 10 and Anaconda.

Suggested Correction:

Lines 22 - 23 states “Histopathological images are routinely used in the diagnosis of many diseases, notably cancer.” This can be misinterpreted as saying that pathologists make their diagnoses predominantly through whole slide images (WSIs). Although WSI is becoming more widespread in pathology departments, most pathologists still render their diagnoses by examining glass slides under a microscope. This statement has to be corrected/modified to reflect that whole slide images are still not being used by majority of pathologists to sign out their cases, although there is an increasing adoption of whole slide scanning technologies in pathology departments.

Future direction:

There is more potential in this software, which can accommodate additional features in the future while retaining its simplicity. Aside from adding new features, I believe adding a Graphical user interface (GUI) version of the program would increase the application’s user base, and be helpful for those who are less computer savvy and have no experience in using the command line.

Reviewer #2: The manuscript submitted by Muñoz-Aguirre and colleagues aims to describe the development

of PyHIST which is a histological image segmentation tool. Overall, this manuscript presents

results that would be of interest to the community of scientists and computational biologists

concerned with this problem. However, there are major issues in this manuscript that prevent us

from recommending that this manuscript be accepted in its current state.

Major:

1) Abstract: highlights that preprocessing enabled by PyHIST involves image scaling,

segmentation, and eventually tile extraction to clearly mention the utility of PyHIST.

2) Introduction: The paper correctly addresses the need for standardization of the tiling and

patch-creating pipeline for researchers working in this area to prevent dataset-specific

biases. Although, as far as saving research time is concerned, currently, WSI preprocessing

requires developing custom scripts, but once a process is established researchers can

typically use similar code for subsequent tiling for all projects. Therefore, PyHIST may only

save a significant amount of time at the initial phase.

3) Facts have been mentioned without references – we have mentioned a few examples but

urge the authors to add extensive references:

- lines 22 (citation for WSI obtaining process required),

- 23 (citation for use in cancer),

- 25 (citation to support the claim of development of computational methods for

disease diagnosis and classification), and

- 33 (cite literature to support histopathological images capturing endophenotypes that

provide crucial information when correlated with molecular and cellular data).

- In a similar way, kindly provide references at lines 37, 46, 50

4) Design and Implementation:

- It’s not clear why the authors are interested in highlighting edges within tissue

fragments rather than outlining the entire fragment. Figure 1b resembles a

grayscaled WSI. A similar result as Figure 1b can be reached with less computation

by just binarizing the WSI using a threshold to separate background from foreground.

Does edge detection provide any unique benefits over binarizing the WSI?

- The graph-based segmentation algorithm can perform unsupervised segmentation

on complex images, but in this case the algorithm just needs to detect the connected

objects. If the input image is a binary mask (foreground and background), there are

many simple functions to label contiguous/interconnected objects and produce an

output similar to Figure 1c. Is graph-based segmentation used because it works well

with edge detection inputs? How does it compare computationally to other

connected-component labeling techniques such as Python Skimage’s measure.label

function?

- Why are steps (b) and (c) needed in the PyHIST pipeline in Figure 1? Red gridlines

still appear to tile the entire WSI and then some tiles are not stored based on a

background threshold. How are the tissue fragment labels from (c) used?

5) Results:

- Details of the deep learning model have not been provided – patches detected

correctly have vague histology that is shown in Figure 2 A (explained below). We

suggest a pathologist review of the deep learning model results. Additionally, the

connection between a better model accuracy on the dataset and the validity of the

pre-processing steps has not been made.

- The partitioning of training and test sets can be the most time-consuming pre-

processing steps of the ML process. Tiles from the same WSI should be constrained

to the training or test sets. It is difficult to satisfy this constraint, while also managing

the percentage of tiles in the test set and class imbalances. This process is not a

built-in feature of PyHIST and it is unclear in the paper if PyHIST assists with this

aspect of the ML pipeline at all.

- The deep learning results are an example that tiles processed using PyHIST can

achieve high prediction performance, but it doesn’t necessarily prove that it is better

than other baseline or competing approaches. WSIs from different part of the body

can be quite distinguishable, so many different tiling approaches could produce

similar results. The Results section could include comparisons of performance and

computation time for several tiling methodologies. How does PyHIST stack up

against other techniques?

6) Availability and future directions:

- The SVS limitation is mentioned here but should also be addressed earlier in the

Intro or Design sections. For example, “PyHIST is currently limited to only SVS

format due to/because…”.

7) Figures:

- Figure 2 A: histology is ambiguous since the top panel for ‘T-brain’ shows artefactual

tissue rather than brain tissue with cell bodies of neurons or glia etc. This is repeated

for 3rd, 4th and 5th (from left) T-breast, and 1st (from left) T-colon.

8) Supplementary Materials:

- Section S3: cropping the image tiles is mentioned – what is the size of these crops

and are these kept uniform each time? Explanation is required for clarity of the user.

- Section S2: the segmentation parameters seem to be an important part of tiling, but it

is still unclear how they work. Is this a way of capturing tiles that have background in

a certain orientation? How do parameters for border and corners interact with the

background percentage and how does this influence segmentation?

**Have all data underlying the figures and results presented in the manuscript been provided?**

Reviewer #1: Yes

Reviewer #2: Yes

PLOS authors have the option to publish the peer review history of their article (what does this mean?). If published, this will include your full peer review and any attached files.

Reviewer #1: **Yes: **Jerome Cheng

Reviewer #2: No
---

## [Decision Letter · Decision Letter 1]

17 Sep 2020

Dear Muñoz-Aguirre,

We are pleased to inform you that your manuscript 'PyHIST: A Histological Image Segmentation Tool' has been provisionally accepted for publication in PLOS Computational Biology.

Best regards,

Dina Schneidman

Software Editor

PLOS Computational Biology

Reviewer's Responses to Questions

**Comments to the Authors:**

Reviewer #1: In the revised and significantly improved version of the manuscript, the authors addressed each reviewer's concerns, and all of my previous comments have been satisfactorily addressed. I do not have any new recommendations.

Reviewer #2: The revised manuscript submitted by Muñoz-Aguirre and colleagues extensively address the comments raised by the reviewers.

We commend them for adding detailed methods regarding pre-processing including tile extraction, additional relevant references, and mask comparisons in Supplementary Text S1 and Supplementary Figure S3. Further the edits done for figure 2 have enabled to message to be clearer and the authors have done a remarkable job.

**Have all data underlying the figures and results presented in the manuscript been provided?**

Reviewer #1: Yes

Reviewer #2: Yes

PLOS authors have the option to publish the peer review history of their article (what does this mean?). If published, this will include your full peer review and any attached files.

Reviewer #1: **Yes: **Jerome Cheng

Reviewer #2: **Yes: **Sana Syed

---

## [Editor Report · Acceptance letter]

9 Oct 2020

PCOMPBIOL-D-20-00862R1 

PyHIST: A Histological Image Segmentation Tool

Dear Dr Muñoz-Aguirre,

I am pleased to inform you that your manuscript has been formally accepted for publication in PLOS Computational Biology. Your manuscript is now with our production department and you will be notified of the publication date in due course.

With kind regards,

Matt Lyles
